# Exogenous Administration of Low-Dose Lipopolysaccharide Potentiates Liver Fibrosis in a Choline-Deficient l-Amino-Acid-Defined Diet-Induced Murine Steatohepatitis Model

**DOI:** 10.3390/ijms20112724

**Published:** 2019-06-03

**Authors:** Keisuke Nakanishi, Kosuke Kaji, Mitsuteru Kitade, Takuya Kubo, Masanori Furukawa, Soichiro Saikawa, Naotaka Shimozato, Shinya Sato, Kenichiro Seki, Hideto Kawaratani, Kei Moriya, Tadashi Namisaki, Hitoshi Yoshiji

**Affiliations:** Third Department of Internal Medicine, Nara Medical University, Kashihara, Nara 634-8521, Japan; nakanishi@naramed-u.ac.jp (K.N.); kitadem@naramed-u.ac.jp (M.K.); taku@naramed-u.ac.jp (T.K.); furukawa@naramed-u.ac.jp (M.F.); saikawa@naramed-u.ac.jp (S.S.); shimozato@naramed-u.ac.jp (N.S.); shinyasato@naramed-u.ac.jp (S.S.); seki@naramed-u.ac.jp (K.S.); kawara@naramed-u.ac.jp (H.K.); moriyak@naramed-u.ac.jp (K.M.); tadashin@naramed-u.ac.jp (T.N.); yoshijih@naramed-u.ac.jp (H.Y.)

**Keywords:** nonalcoholic steatohepatitis, lipopolysaccharide, choline-deficient l-amino-acid-defined diet, toll-like receptor, NF-κB, fibrosis

## Abstract

Various rodent models have been proposed for basic research; however, the pathogenesis of human nonalcoholic steatohepatitis (NASH) is difficult to closely mimic. Lipopolysaccharide (LPS) has been reported to play a pivotal role in fibrosis development during NASH progression via activation of toll-like receptor 4 (TLR4) signaling. This study aimed to clarify the impact of low-dose LPS challenge on NASH pathological progression and to establish a novel murine NASH model. C57BL/6J mice were fed a choline-deficient l-amino-acid-defined (CDAA) diet to induce NASH, and low-dose LPS (0.5 mg/kg) was intraperitoneally injected thrice a week. CDAA-fed mice showed hepatic CD14 overexpression, and low-dose LPS challenge enhanced TLR4/NF-κB signaling activation in the liver of CDAA-fed mice. LPS challenge potentiated CDAA-diet-mediated insulin resistance, hepatic steatosis with upregulated lipogenic genes, and F4/80-positive macrophage infiltration with increased proinflammatory cytokines. It is noteworthy that LPS administration extensively boosted pericellular fibrosis with the activation of hepatic stellate cells in CDAA-fed mice. Exogenous LPS administration exacerbated pericellular fibrosis in CDAA-mediated steatohepatitis in mice. These findings suggest a key role for LPS/TLR4 signaling in NASH progression, and the authors therefore propose this as a suitable model to mimic human NASH.

## 1. Introduction

Nonalcoholic fatty liver disease (NAFLD) is defined as ectopic lipid accumulation in the liver without other secondary causes such as chronic viral hepatitis, excessive alcohol consumption, autoimmune hepatitis, or congenital hepatic disorders [1,2]. NAFLD patients often develop nonalcoholic steatohepatitis (NASH), which is histologically characterized by the presence of hepatic inflammation and may potentially progress to fibrosis, cirrhosis, and eventually hepatocellular carcinoma (HCC) [3,4]. NASH is a currently growing cause of liver cirrhosis and HCC in developed countries, and a recent longitudinal study in NASH patients showed that the fibrosis stage is uniquely associated with long-term overall mortality, liver transplantation, and liver-related events [5,6]. However, effective pharmacotherapy targeting NASH-based fibrosis has not been established yet. Therefore, further experimental studies using animal models pathologically resembling the human NASH phenotype are required to establish novel therapeutic approaches.

Several animal models are currently being used to mimic human NASH. In particular, the effect of choline deficiency on NASH development is well established, corroborated by the fact that a choline-deficient l-amino-acid-defined (CDAA) diet is frequently utilized in murine NASH models [7,8,9]. Although these CDAA-fed rodents are available as NASH experimental models, they are not entirely inadequate for mimicking NASH, as some important characteristics are missing. For example, a CDAA-fed rat lacks obesity and insulin resistance (IR) [10]. Moreover, CDAA-fed mice exhibiting obesity and IR develop limited liver fibrosis [11]. This defect is inconvenient for NASH fibrogenesis basic research and evidences the need for a further improved CDAA-fed mouse model.

NASH pathogenesis is often interpreted with the multiple parallel hits hypothesis, and various pathways have been recently implicated [12]. Among these, the toll-like receptor (TLR) 4 signaling pathway is one of the most relevant pathways involved in NASH onset and progression. It is well acknowledged that the TLR4 pathway is activated by its predominant ligand, lipopolysaccharide (LPS), a component of several gram-negative and selected gram-positive bacteria [13]. Indeed, NASH patients usually display bacterial overgrowth or intestinal hyperpermeability, and LPS is acknowledged as a major inducer of neutrophil infiltration in NASH patients’ liver [14,15,16]. The activated TLR4 signaling pathway leads to NF-κB intracellular signal transduction and inflammatory cytokine production responsible for mediating the innate immune response [17]. In the liver, TLR4 is normally expressed in various cell types, including hepatocytes, Kupffer cells (KCs), and hepatic stellate cells (HSCs), and plays a key role in the linkage between intestinal microbiota, endotoxemia, and liver damage [18]. By applying this molecular basis, we have recently shown that fructose administration increased the endogenous LPS overload in the liver, resulting in marked liver fibrosis exacerbation and hepatocarcinogenesis with induction of LPS/TLR4 signaling in a CDAA-fed rat model [19]. Although this model provides mechanistic evidence of the interaction between increased LPS and NASH progression, it is not completely suitable as a human NASH-mimicking model, as it exhibits remarkable weight loss and lacks IR. Furthermore, it remains unclear how exogenous LPS administration influences CDAA-mediated NASH pathogenesis.

In this study, we aimed to (i) clarify the impact of LPS challenge on the pathology of NASH fibrosis by applying repetitive intraperitoneal low-dose LPS injections on CDAA-fed mice and (ii) set the foundation for establishing a more suitable human NASH-mimicking model.

## 2. Results

### 2.1. Repetitive Low-Dose LPS Infusion Triggers Activation of the TLR4/NF-κB Signaling Pathway in the Liver of CDAA-Fed Mice

The experimental protocols used in this study are outlined in Figure 1A. To evaluate LPS hepatic exposure via intraperitoneal infusion, hepatic *Lbp* mRNA levels were initially measured. *Lbp* binds to LPS to form a complex that interacts with the macrophage receptor, eliciting the host proinflammatory response. As shown in Figure 1B, hepatic *Lbp* expression was significantly higher in both LPS infusion groups. The mRNA levels of *Cd14*, a co-receptor of TLR4 for LPS detection, were overexpressed in the liver of CDAA-fed mice, and LPS infusion markedly enhanced such *Cd14* overexpression (Figure 1C). The impact of LPS infusion on the LPS/TLR4 signaling pathway was subsequently assessed. In CDAA-fed mice, hepatic TLR4 expression was increased in accordance with increased CD14 expression; notably, additive LPS infusions prominently upregulated the hepatic TLR4 expression in CDAA-fed mice (Figure 1D,E). This LPS-mediated TLR4 upregulation led to enhanced NF-κB activation, detected by its phosphorylation (Figure 1D,F). Additionally, LPS infusion did not substantially change LPS/TLR4 signal transduction activity in a choline-supplemented amino acid-defined (CSAA)-fed mice.

### 2.2. Exogenous LPS Exacerbates Steatosis in CDAA- but not in CSAA-Fed Mice

The body weight of CDAA-fed mice remained unchanged as compared with that of CSAA-fed mice. Additionally, LPS infusion did not affect the body weight of CSAA- or CDAA-fed mice (Figure 2A). It is noteworthy that the CDAA diet alone did not change liver weight; however, LPS infusion significantly increased it in CDAA-fed mice (Figure 2B). These results suggest that the additive administration of LPS causes hepatomegaly without inducing obesity. Histological findings through hematoxylin and eosin (H&E) staining indicated hepatic steatosis in CDAA-fed mice, and LPS overload remarkably augmented hepatic fat accumulation only in CDAA-fed mice (Figure 2C). In accordance with altered histological features, the alanine aminotransferase (ALT) and triglyceride (TG) serum levels were increased following CDAA diet and LPS administration (Figure 2D). The total cholesterol (T-Cho) serum level was unchanged after LPS administration, suggesting that LPS infusion may contribute to fatty acid synthesis.

### 2.3. Alterations in Glucose and Lipid Metabolism Related to CDAA Diet and Low-DOSE LPS Administration

To further explore the underlying mechanism of steatohepatitis, alterations in glucose and lipid metabolism following CDAA diet and LPS administration were examined. At the end of the experiment, an oral glucose tolerance test (OGTT) was performed to determine the differential glycemic status among the experimental groups, with OGTT showing that the CDAA diet impaired glucose tolerance (Figure 3A). Plasma glucose’s area under the curve estimate in OGTT indicated that CDAA-fed groups exhibited hyperglycemia, with a significant difference compared with CSAA-fed groups (Figure 3B). In keeping with glucose intolerance, CDAA-fed groups also induced hyperinsulinemia (Figure 3C). To evaluate IR status, the multiplication of glucose and insulin levels was calculated, with CDAA-fed mice showing significantly higher IR than CSAA-fed mice (Figure 3D). Note that LPS infusion exclusively impaired IR status in CDAA-fed mice (Figure 3C,D).

The impact of CDAA diet and/or LPS administration on hepatic lipid metabolism was subsequently assessed. In CSAA-fed mice, LPS administration did not alter the hepatic expression of lipogenesis-related genes, including *Srebf1*, *Fas*, and *Acc1*, whereas CDAA diet increased the expression of these genes (Figure 3E). Importantly, low-dose LPS infusion induced a marked upregulation of the referred lipogenic genes in CDAA-fed mice (Figure 3E). Regarding fatty acid oxidation, hepatic *Ppara* mRNA levels significantly decreased only in CDAA-fed and LPS-infused mice (Figure 3F). This suggests that LPS administration may accelerate de novo hepatic lipogenesis and attenuate fatty acid oxidation by the influence of hepatic lipid accumulation. Moreover, upregulation of the adipogenic gene *Pparg* was identified in CDAA-fed and LPS-infused mice, in agreement with hepatic steatosis (Figure 3G).

### 2.4. Exogenous LPS Promotes Hepatic Macrophage Infiltration and Inflammatory Response in CDAA-Fed Mice

Given the steatosis enhancement following LPS additive administration, the inflammatory status in each experimental group was evaluated. As shown in Figure 4A, F4/80-positive macrophages mildly infiltrated the liver of CDAA-fed mice, and LPS infusion profoundly accelerated hepatic macrophage infiltration in CDAA-fed mice; however, it had a negligible effect on CSAA-fed mice. Semiquantitative analysis showed a threefold increase in F4/80-positive cells in the CDAA-fed and LPS-infused groups as compared with the CDAA-fed-only group (Figure 4B). In this setting, hepatic mRNA levels of proinflammatory cytokines, including *Tnfa*, *Il1b*, and *Il6*, were limitedly increased in the CDAA-diet-only group, whereas they were substantially increased in the CDAA-fed and LPS-infused group (Figure 4C). Therefore, LPS challenge apparently facilitates steatohepatitis progression by enhanced hepatic macrophage infiltration and inflammatory response.

### 2.5. Exogenous LPS Robustly Boosts Liver Fibrosis Development in CDAA-Fed Mice

Consistently with exacerbated steatohepatitis, CDAA diet induced liver fibrosis development, and LPS infusion markedly enhanced CDAA-diet-induced liver fibrosis (Figure 5A). Semiquantitative analysis showed a fourfold increase in liver fibrotic area in CDAA-fed and LPS-administered mice as compared with CDAA-fed-only animals (Figure 5B). Immunohistochemical analysis of α-smooth muscle actin (α-SMA) staining was then performed to assess the activation of HSCs, which are known to play a crucial role in hepatic fibrogenesis. It was observed that CDAA-fed mice who were also administered LPS showed a 17-fold increase in α-SMA-positive areas compared with CDAA-fed-only mice (Figure 5A,C). This fibrotic progression coincided with the increased hepatic expression of profibrotic genes, including *Acta2*, *Col1a1,* and *Tgfb1* (Figure 5D). These findings indicate that additive LPS infusion substantially boosts liver fibrosis more than a CDAA diet alone.

## 3. Discussion

To generate a murine model mimicking human NASH, the present study proposed modifying the CDAA-fed mice model—a well-established model for steatohepatitis induction—with additional low-dose LPS challenge. CDAA-fed mice typically show mild liver steatosis, inflammation, and fibrosis, and low-dose LPS administration markedly exacerbated these pathological phenotypes. The results from the present study initially showed that a CDAA diet significantly induced LR4/NF-κB signaling activation following hepatic CD14 overexpression. This is in line with our previous findings showing that KCs isolated from CDAA-fed rats displayed higher CD14 expression than those from pair diet-fed rats [20]. It is noteworthy that as per a recent report by Imajo et al., high-fat-diet (HFD)-mediated hyperleptinemia could induce CD14 overexpression through STAT3 signaling activation in KCs [21]. Similarly to HFD, the CDAA diet reportedly increased the serum leptin levels, suggesting that increased CD14 expression in CDAA-fed mice possibly also triggers LPS hyperresponsivity through STAT3 activation in KCs [22].

In the current model, administration of low-dose LPS potentiated hepatic steatosis with impaired IR in CDAA-fed mice, indicating that low-dose LPS infusion could induce de novo hepatic lipogenesis. Interestingly, unlike CDAA-fed rats, CDAA-fed mice did not show any significant weight loss. In agreement with this result, a recent report has shown that CDAA-fed C57BL/6J mice gained as much or more weight than mice on a standard diet [14]. Moreover, several experimental data suggest an interaction between LPS/TLR4 and hepatic lipid accumulation and that inactivation of TLR4 attenuates hepatic steatosis [23]. As per a recent report by Yang et al., in this study hepatocytes in CDAA-fed mice developed steatosis via the TLR4/TRIF pathway and independently of the TLR4/Myd88 pathway, strongly supporting the LPS-mediated potentiation of hepatic steatosis in the same model [24]. As expected, LPS infusion enhanced CDAA-diet-induced hepatic infiltration of F4/80-positive macrophages and inflammatory response, as indicated by the upregulation of proinflammatory cytokines. Miura et al. reported that CDAA-fed mice demonstrated increased hepatic recruitment of Ly6C^+^CD68^+^CCR2^+^ bone-marrow-derived macrophages [25]. On the basis of this evidence, we also speculate that increased macrophages in CDAA-fed and LPS-infused mice are derived from the bone marrow. However, further investigation is required to elucidate their true origin. We emphasize that CDAA-fed mice show pronounced liver fibrosis development following low-dose LPS administration, in accordance with KC-mediated inflammatory response. It is noteworthy that this LPS challenge exacerbated the development of liver fibrosis to a greater extent than macrophage infiltration. A great deal of evidence proposes that HSCs also play a crucial role in promoting fibrosis in a TLR4-dependent manner. Seki et al. demonstrated that both quiescent and activated HSCs exhibit high levels of TLR4 expression and that LPS directly targets HSCs [26]. Moreover, TLR4 activation downregulates Bambi, a TGF-β pseudo-receptor, and sensitizes HSCs to TGF-β [26]. These findings strongly corroborate that low-dose LPS challenge directly affected HSCs in this study, in addition to sequential induction by enhanced inflammatory response.

This study has several limitations that deserve consideration. First, it only analyzed male mice. Sex difference is a definite feature of NAFLD, and sexual dimorphism in NAFLD is increasingly recognized [27,28,29,30,31,32]. In fact, a recent large cohort study showed that decreased choline intake is significantly associated with increased NASH fibrosis, which was remarkable in postmenopausal women [33]. Moreover, the response to the LPS inflammation challenge is reportedly affected by mouse sex differences [34]. All this evidence requires further investigation to evaluate whether CDAA/LPS exerts a similar effect in female mice. Second, housing temperature is also critical for LPS-mediated NASH progression. Giles et al. showed that thermoneutral housing at 30 ºC exacerbated NAFLD pathogenesis during HFD feeding, with altered immune responsiveness by norepinephrine and corticosterone decrease [35]. For further clarification of murine NASH pathogenesis, it is also important to examine how thermoneutral housing affects the proposed model. Moreover, several translational strategies need to be developed and explored to expand our basic findings to the clinical field. As previously shown, decreased choline intake is involved in fibrosis promotion in NASH patients, suggesting that choline plasma levels could be a biomarker to evaluate NASH severity [33]. It is also important to clinically investigate whether LPS pharmacological reduction—by nonabsorbable antibiotics, for example—inhibits fibrosis development in NASH patients with decreased choline intake.

To the best of our knowledge, this is the first report showing that low-dose LPS administration has a direct detrimental effect on CDAA-mediated NASH in mice. Although we demonstrate that additive LPS overload deteriorates each process during NASH progression—including steatosis, inflammation, and fibrosis development—detailed molecular approaches are required to elucidate the functional role of LPS/TLR4 signaling in pathological status in the present model. Given that both CDAA diet and LPS infusion are simple and convenient, this method may be a legitimate option to adjust a well-established model mimicking human NASH.

## 4. Materials and Methods

### 4.1. Animals and Experimental Protocol

Six-week-old male C57BL/6J mice (CLEA Japan, Inc., Osaka, Japan) were divided into the following four groups: (1) CSAA diet (Research Diets, Inc., New Brunswick, NJ, USA) with intraperitoneal injection (i.p.) of vehicle (CSAA group, *n* = 6); (2) CSAA diet with i.p. of low-dose LPS (0.5 mg/kg, thrice a week) (CSAA + LPS group, *n* = 6); (3) CDAA diet (Research Diets, Inc.) with i.p. of vehicle (CDAA group, *n* = 6); and (4) CDAA diet with i.p. of low-dose LPS (0.5 mg/kg, thrice a week) (CDAA + LPS group, *n* = 6). LPS (from *E. coli* O55, Wako, Osaka, Japan) was administered via i.p. through the McBurney’s point in mice under isoflurane-inhaled anesthesia. Mice in the vehicle group were administered an equivalent volume of saline solution. All the mice were sacrificed after 16 breeding weeks. At the end of the experiment, all the mice were orally administered 2 g/kg of glucose, and the plasma glucose levels were measured after 0, 15, 30, 60, and 120 min in OGTT. Mice were anesthetized, their abdominal cavities were opened, blood samples were drawn via aortic puncture, and the livers were harvested for histological examination. Serum biological markers were assessed using routine laboratory methods, and insulin levels were measured using the ELISA kit (Mercodia, Uppsala, Sweden). Mice were housed in stainless steel mesh cages under controlled conditions (temperature: 23 °C ± 3 °C; relative humidity: 50% ± 20%; 10–15 air changes/h; illumination: 12 h/day). Animals were allowed access to tap water ad libitum throughout the experimental period. All animal procedures were performed in compliance with the recommendations of the Guide for Care and Use of Laboratory Animals (National Research Council), and the study was approved by the animal facility committee of Nara Medical University (http://www.naramed-u.ac.jp/university/kenkyu-sangakukan/sentanigaku/dobutsujikken/dobutsujikken.html) (Code: 11757, date of approval: 2 August 2016).

### 4.2. Histological and Immunohistochemical Analyses

Liver sections were fixed with 10% formalin and embedded in paraffin. Subsequently, 5-µm paraffin sections were stained with H&E and Sirius Red. α-SMA (Abcam, Cambridge, UK), and F4/80 (Abcam) primary antibodies were used for immunostaining, with staining performed as per the supplier’s recommendations. For quantitative analysis, NIH ImageJ software (http://imagej.nih.gov/ij/) was used.

### 4.3. RNA Extraction and Reverse Transcription Quantitative Polymerase Chain Reaction (RT-qPCR)

Total RNA was extracted from frozen liver tissues using the RNeasy mini kit (QIAGEN, Tokyo, Japan) as per the manufacturer’s instructions. The total RNA from each sample was subsequently reverse transcribed into complementary DNA (cDNA) using the high-capacity RNA-to-cDNA kit (Applied Biosystems Inc., Foster City, CA, USA) as per the manufacturer’s instructions. cDNA RT-qPCR was performed using gene-specific primer pairs (Appendix A) and Step One Real Time PCR (Applied Biosystems, Foster City, CA, USA). Relative gene expression was measured using glyceraldehyde-3-phosphate dehydrogenase as an internal control. The relative amount of target mRNA per cycle was determined by applying a threshold cycle to the standard curve.

### 4.4. Protein Extraction and Western Blotting

Whole-cell lysates were prepared from 200 mg of frozen liver tissue using a T-PER Tissue Protein Extraction Reagent supplemented with proteinase and phosphatase inhibitors (all Thermo Scientific, Rockford, IL, USA). In total, 50 μg of whole-cell lysates were separated with SDS-PAGE (NuPAGE, Thermo Fisher Scientific Inc., Waltham, MA, USA) and transferred to an Invitrolon PVDF membrane (Thermo Fisher Scientific Inc.) that was subsequently blocked with 5% bovine serum albumin in Tris-buffered saline + Tween-20 (TBS-T) for 1 h. Each membrane was then incubated overnight with a TLR4-directed antibody, NF-κB and phosphorylated NF-κB (Cell Signaling Technology, Danvers, MA, USA), and β-actin (Merck, Darmstadt, Germany). Membranes were then washed and incubated with Amersham ECL IgG, HRP-linked F(ab)2 fragment (GE Healthcare Life Sciences, Piscataway, NJ, USA; 1:5000 dilution). Finally, each membrane was developed using Clarity Western ECL Substrate (BIORAD, Hercules, CA, USA).

### 4.5. Statistical Analysis

Retrieved data were subjected to the Student’s *t*-test or one-way analysis of variance followed by Bonferroni’s multiple-comparison test, as appropriate. Bartlett’s test was used to determine the homology of variance. Statistical analyses were performed using Prism, version 6.04 (GraphPad Software, La Jolla, CA, USA). All tests were two-tailed, and *p*-values < 0.05 were considered to be statistically significant.

## Figures and Tables

**Figure 1 ijms-20-02724-f001:**
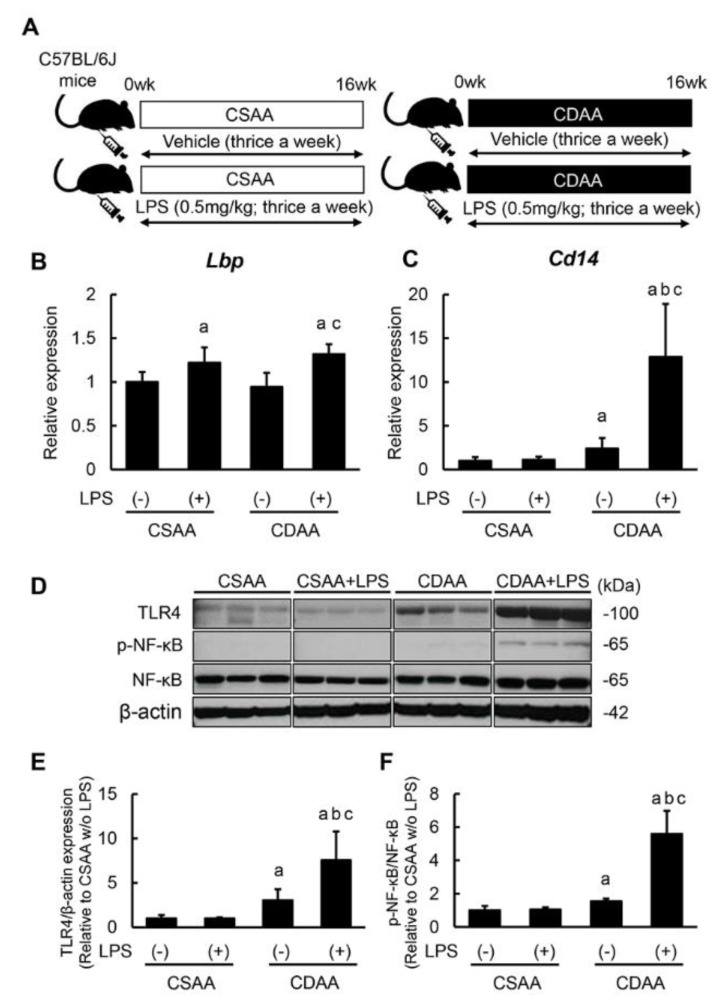
Activation of hepatic TLR 4/NF-κB signaling pathway by CDAA diet feeding and LPS administration. (**A**) Schematic of LPS administration to CDAA diet-induced steatohepatitis model. (**B**,**C**) Relative mRNA expression levels of *Lbp* (**B**) and *Cd14* (**C**) in the liver of experimental mice. (**D**) Western blots for TLR4 expression and NF-κB phosphorylation in the liver of experimental mice. (**E**) Quantification of protein expression of TLR4. (**F**) Quantitative phosphorylation rate of phosphorylated NF-κB/NF-κB. The mRNA expression levels were measured by quantitative RT–PCR (qRT–PCR), and *Gapdh* was used as internal control for qRT–PCR (**B**,**C**). The protein was determined by western blotting, and β-Actin was used as the loading control (**D**). Quantitative values are indicated as ratios to the values of CSAA-LPS (−) group (**B**,**C**,**E**,**F**). Data are mean ± SD (*n* = 6). ^a^: *p* < 0.05 compared with CSAA-LPS (−), ^b^: *p* < 0.05 compared with CSAA-LPS (+), ^c^: *p* < 0.05 compared with CDAA-LPS (−).

**Figure 2 ijms-20-02724-f002:**
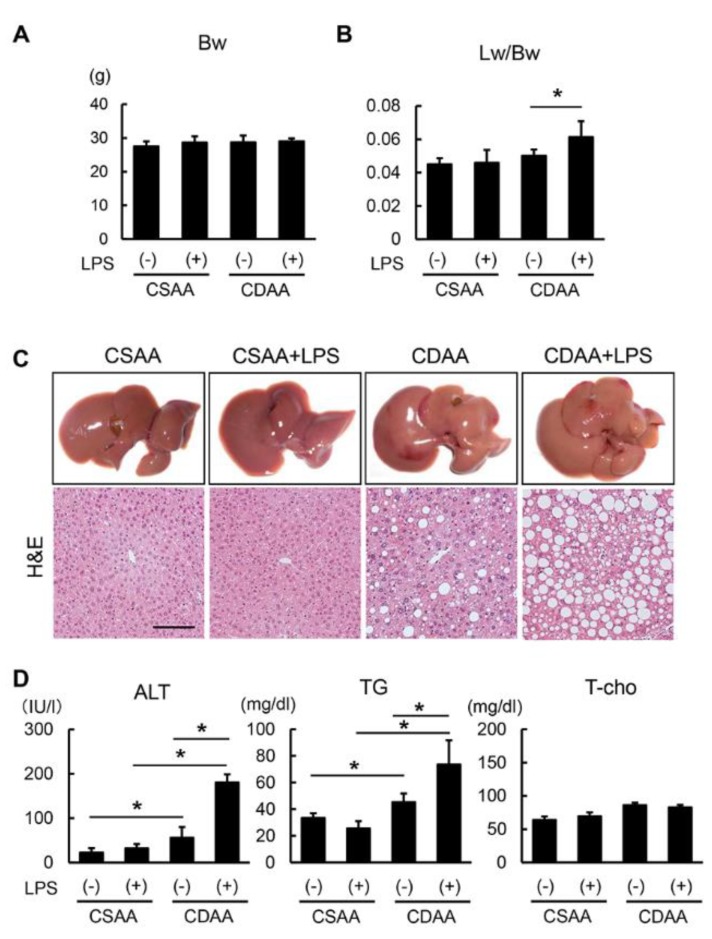
Altered characteristics and hepatic steatosis by CDAA diet feeding and LPS administration. (**A**) Body weight (Bw) in the experimental groups at sacrifice. (**B**) Ratio of liver weight (Lw) to body weight in the experimental groups at sacrifice. (**C**) Representative macroscopic appearances and microphotographs of hematoxylin-eosin (H&E) staining in the experimental groups. Scale Bar: 50 μm. (**D**) Serum levels of alanine aminotransferase (ALT), triglyceride (TG) and total cholesterol (T-Cho) in the experimental groups. Data are mean ± SD (*n* = 6). * *p* < 0.05, indicating a significant difference between groups.

**Figure 3 ijms-20-02724-f003:**
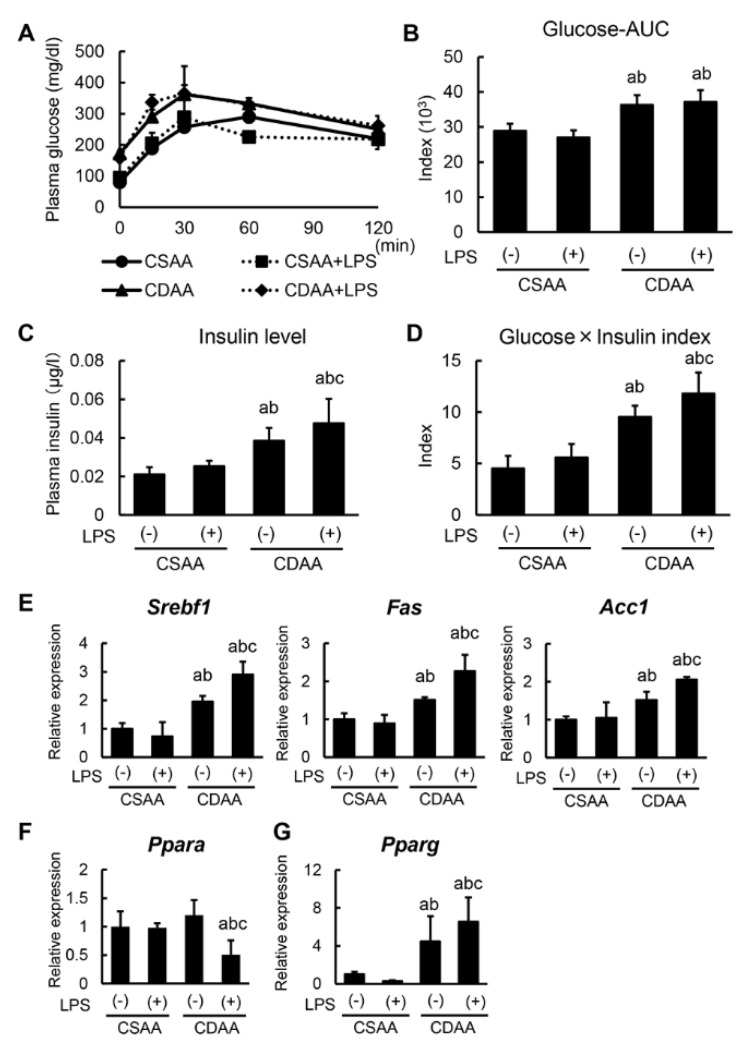
Changes in glycemic status and lipid metabolism by CDAA diet feeding and LPS administration. (**A**) Plasma glucose levels after 0, 15, 30, 60, 120 min for oral glucose tolerance test (OGTT) at the end of experiment. (**B**) The values of Glucose-AUC (area under the blood concentration-time curve) in the experimental group. The value of AUC was calculated as the area under the curve of plasma glucose level at OGTT. (**C**) Plasma insulin levels in the experimental groups at sacrifice (120 min. after oral glucose intake). (**D**) The multiplicated values of plasma glucose and insulin level in the experimental groups at sacrifice (120 min. after oral glucose intake). (**E**–**G**) Relative mRNA expression levels of *Srebf1*, *Fas*, *Acc1* (**E**), *Ppara* (**F**) and *Pparg* (**G**) in the liver of experimental mice. The mRNA expression levels were measured by quantitative RT–PCR (qRT–PCR), and *Gapdh* was used as internal control for qRT–PCR (**E**–**G**). Quantitative values are indicated as ratios to the values of CSAA-LPS (−) group (**E**–**G**). Data are mean ± SD (*n* = 6). ^a^: *p* < 0.05 compared with CSAA-LPS (−), ^b^: *p* < 0.05 compared with CSAA-LPS (+), ^c^: *p* < 0.05 compared with CDAA-LPS (−).

**Figure 4 ijms-20-02724-f004:**
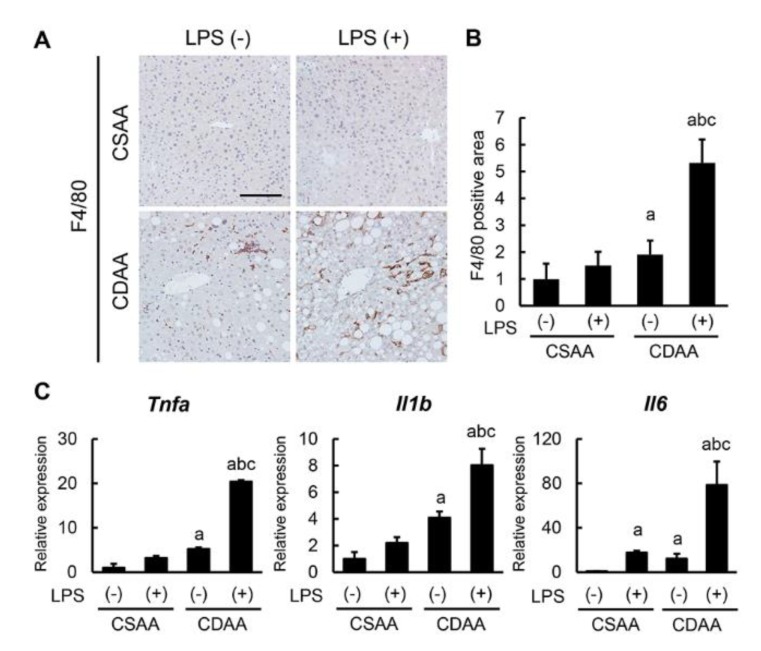
Impact of CDAA diet feeding and LPS administration on macrophage infiltration and inflammatory response. (**A**) Representative microphotographs of liver sections stained with F4/80. Scale bar: 50 μm. (**B**) Semi-quantitation of F4/80 immuno-positive macrophages in high-power field (HPF) by NIH imageJ software. (**C**) Relative mRNA expression levels of proinflammatory cytokines, *Tnfa*, *Il1b* and *Il6* in the liver of experimental mice. The mRNA expression levels were measured by quantitative RT–PCR (qRT–PCR), and *Gapdh* was used as internal control for qRT–PCR (**C**). Quantitative values are indicated as ratios to the values of CSAA-LPS (−) group (**B**,**C**). Data are mean ± SD (*n* = 6). ^a^: *p* < 0.05 compared with CSAA-LPS (−), ^b^: *p* < 0.05 compared with CSAA-LPS (+), ^c^: *p* < 0.05 compared with CDAA-LPS (−).

**Figure 5 ijms-20-02724-f005:**
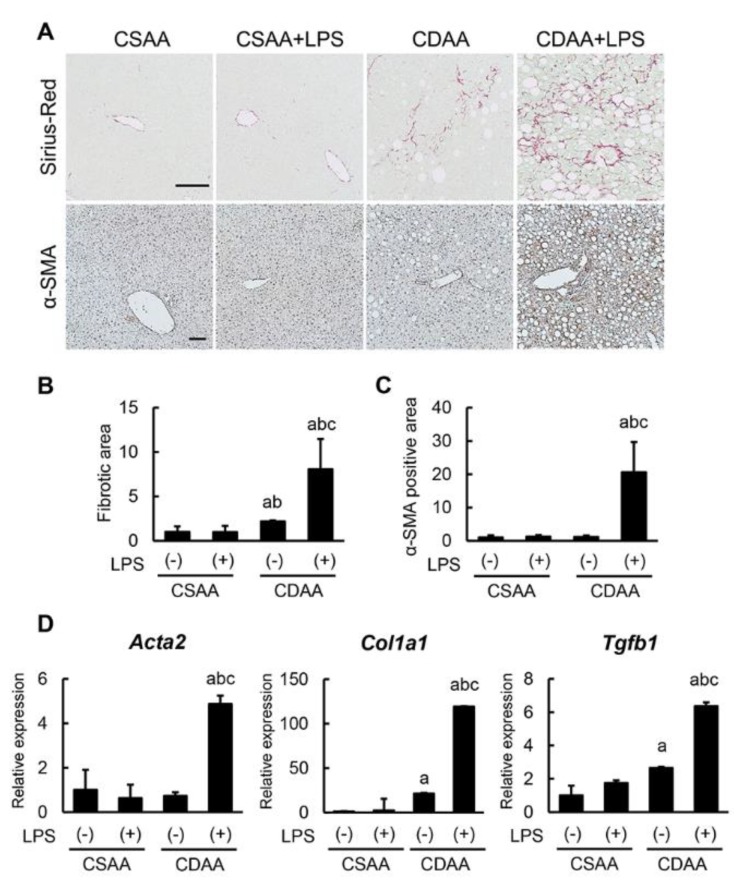
Impact of CDAA diet feeding and LPS administration on liver fibrosis development. (**A**) Representative microphotographs of liver sections stained with Sirius Red (upper panels) and α-SMA (lower panels). Scale Bar: 50 μm. (**B**,**C**) Semi-quantitation of Sirius Red-stained fibrotic area (**B**) and α-SMA immune-positive area (**C**) in high-power field (HPF) by NIH imageJ software. (**D**) Relative mRNA expression levels of fibrosis markers, *Acta2*, *Col1a1* and *Tgfb1* in the liver of experimental mice. The mRNA expression levels were measured by quantitative RT–PCR (qRT–PCR), and *Gapdh* was used as internal control for qRT–PCR (**D**). Quantitative values are indicated as ratios to the values of CSAA-LPS (−) group (**C**,**D**). Data are mean ± SD (*n* = 6). ^a^: *p* < 0.05 compared with CSAA-LPS (−), ^b^: *p* < 0.05 compared with CSAA-LPS (+), ^c^: *p* < 0.05 compared with CDAA-LPS (−).

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
