# Peer review of "Exogenous Administration of Low-Dose Lipopolysaccharide Potentiates Liver Fibrosis in a Choline-Deficient l-Amino-Acid-Defined Diet-Induced Murine Steatohepatitis Model"

_ijms, 2019, doi:10.3390/ijms20112724_

Round 1

Reviewer 1 Report

This is a nice paper describing the impact of low doses of exogenous LPS on the development of liver fibrosis in a CDAA diet-induced murine steatohepatitis model. In this model, LPS exacerbated pericellular fibrosis, confirming the role of LPS/TLR4 signaling in NASH progression.

Although not major breakthrough is reported, the originality comes from the combination of low doses of LPS with a CDAA diet. I anticipate that this approach will be useful for further NAFLD/NASH exploration in the authors' and other laboratories.

The work is well done and there is no major comment about it. However, the molecular basis for the LPS/CDAA collaborative effect is not yet elucidated (see last paragraph of the Discussion), but one can consider that this would go beyond the scope of this work.

In spite of the fact that the paper has been reviewed for English language, the English is not satisfactory and must be improved. It will make this nice paper more readable.

Author Response

Reviewer 1.

We thank the Reviewer for his/her positive evaluation of our work.

This is a nice paper describing the impact of low doses of exogenous LPS on the development of liver fibrosis in a CDAA diet-induced murine steatohepatitis model. In this model, LPS exacerbated pericellular fibrosis, confirming the role of LPS/TLR4 signaling in NASH progression.

Although not major breakthrough is reported, the originality comes from the combination of low doses of LPS with a CDAA diet. I anticipate that this approach will be useful for further NAFLD/NASH exploration in the authors' and other laboratories.

The work is well done and there is no major comment about it. However, 1) the molecular basis for the LPS/CDAA collaborative effect is not yet elucidated (see last paragraph of the Discussion), but one can consider that this would go beyond the scope of this work.

In spite of the fact that the paper has been reviewed for English language, the 2) English is not satisfactory and must be improved. It will make this nice paper more readable.

Answer

We greatly appreciate the reviewer’s comment.

1) As the reviewer stated, the detailed molecular basis of the LPS/CDAA collaborative effect still remain unclear in this manuscript. Yang et al have shown that CDAA diet affects the composition and the amount of intestinal bacteria and promotes TLR4 activation by endogenous LPS in Kupffer cells (KCs), hepatic stellate cells (HSCs) and hepatocytes. They have elucidated the complex mechanism of promoted steatohepatitis in CDAA-fed mice, showing that activated TLR4 subsequently activates MyD88-dependent and TRIF-dependent pathways. KCs produce CCL3 and CXCL1 through the MyD88-dependent pathway and produce CCL5 and IL10 through the TRIF-dependent pathway. In hepatocytes, CCL3 and CXCL1 production is MyD88-dependent whereas CCL5 induction is TRIF-dependent. In contrast, all CCL3, CCL5, and CXCL1 induction and Bambi downregulation is mediated through MyD88 in HSCs (Cell Mol Gastroenterol Hepatol. 3:469-483. 2017). We speculate that these molecular events are boosted by additive administration of LPS.

Now, we are accumulating the data of inflammatory and fibrogenic potentials and TLR4 signaling activation in KCs, HSCs and hepatocytes isolated from LPS+CDAA-mediated mice, and we will show these data in further report.

2) As the reviewer’s comment, this manuscript has got the English proofreading again to be refined. Therefore, we changed the title “Exogenous administration of low-dose lipopolysaccharide potentiates liver fibrosis in a choline-deficient l-amino-acid-defined diet-induced murine steatohepatitis model”.

Reviewer 2 Report

SPECIFIC COMMENT

The development and validation of novel NASH animal models is a challenging  topic. The optimal model should cost as least animal suffering as possible. In this setting the present submission, although has a robust biologial rationale, is far from ideal. In addition, the manuscript must be reworked by discussing certain specific features pertaining to the experimental protocol such as sexual dimorphism of disease as wel as temperature of housing. Finally, the discussion reads such as a review of the literature, is too long and poorly focused on novel findings. Expand the translational potential of the study.    

GENERAL COMMENT

I think that, as researchers, we should constantly aim at a sparing experimental animals unnecessary sufferings. From this perspective, a model featuring repeated intraperitoneal infusions is distant from the ideal. Could these Authors comment on this important ethical topic ?

Background "limited data have been reported on the pathophysiological influence of exogenous LPS administration on the aggravation of NASH, especially liver fibrosis development". It seems to me that this section fails to sufficiently highlight the RATIONALE of the study. Stated otherwise what (based on background information) did these authors expect to find and why ?

Experimental protocol

Sex differences are a definite feature of NAFLD and sexual dimorphism in NAFLD is increasingly being recognized. These authors used male animals. Could this be a limitation of their study ? Please discuss this topic based on the most recently published evidence.

Along the same line, these authors used a temperature for housing animals which is probably not the most physiological to use (Nat Med. 2017 Jul;23(7):829-838). Could dthese Authors comment on this methodological aspect ?

The discussion is too long and poorly focused on novel findings. For example Lines 145-162 should better be deleted (This is a topic to be addressed in the Background section). Rework the layout of this section into a four-paragraph structure as follows: a) Start the discussion by summarizing novel findings. b) Next, discuss them in the setting of previosu study. c) List limitations and strengths of the model. d) Finally, propose a research agenda specifically based on novel findings.

Expand the translational potential of findings, as much as possible. For example, it must be highlighted that decreased choline intake is significantly associated with increased fibrosis in postmenopausal women with NAFLD (Am J Clin Nutr. 2012;95:892-900) which, again, raised the topic of sex differences in NAFLD.

Author Response

Reviewer 2.

We thank the Reviewer for his/her positive evaluation of our work.

SPECIFIC COMMENT

The development and validation of novel NASH animal models is a challenging topic. The optimal model should cost as least animal suffering as possible. In this setting the present submission, although has a robust biologial rationale, is far from ideal. In addition, the manuscript must be reworked by discussing certain specific features pertaining to the experimental protocol such as sexual dimorphism of disease as well as temperature of housing. Finally, the discussion reads such as a review of the literature, is too long and poorly focused on novel findings. Expand the translational potential of the study.    

GENERAL COMMENT

I think that, as researchers, we should constantly aim at a sparing experimental animals unnecessary sufferings. From this perspective, a model featuring repeated intraperitoneal infusions is distant from the ideal. Could these Authors comment on this important ethical topic?

Answer

We agree with the reviewer’s comment that repeated intraperitoneal injection cause mice suffering. Initially, we attempted to orally administer LPS to CDAA-fed mice to generate a novel NASH model in our preliminary experiment. However, it failed to produce a phenotype mimicking human NASH in this model, because it was difficult to administer in equal amounts of LPS to every mouse. Therefore, we necessarily employed intraperitoneal injection as route of administration. Actually, we performed intraperitoneal injection under anesthesia with isoflurane inhalation to alleviate their pain every time.

We apologize the reviewer for our insufficient description missing “under anesthesia”. We added this important description in Materials and Methods of the revised manuscript (line 214-215).

Background "limited data have been reported on the pathophysiological influence of exogenous LPS administration on the aggravation of NASH, especially liver fibrosis development". It seems to me that this section fails to sufficiently highlight the RATIONALE of the study. Stated otherwise what (based on background information) did these authors expect to find and why?

Answer

We agree with the reviewer’s comment and apologize for insufficient description. Eventually, the goal of this study is to briefly generate a NASH model which rapidly develops fibrosis for short experimental period. CDAA (+fructose)-fed rat model is suitable this purpose, but this model shows remarkable weight loss being inappropriate for mimicking human NASH. On the other hand, CDAA-fed mouse model does not show weight loss, but liver fibrosis development is comparatively mild. Therefore, we expected that additive LPS administration to CDAA-fed mouse can approach the more ideal model by augmentation of liver fibrosis development via promoted TLR4 activation. We correctly changed this part in Introduction of the revised manuscript (line 64-70).

Experimental protocol

Sex differences are a definite feature of NAFLD and sexual dimorphism in NAFLD is increasingly being recognized. These authors used male animals. Could this be a limitation of their study? Please discuss this topic based on the most recently published evidence.

Along the same line, these authors used a temperature for housing animals which is probably not the most physiological to use (Nat Med. 2017 Jul;23(7):829-838). Could these Authors comment on this methodological aspect?

Answer

We appreciate the reviewer’s suggestion. As the reviewer stated, sex difference is a definite feature of NAFLD. Additionally, recent report has shown that recovery from an acute systemic and central LPS-inflammation challenge is affected by mouse sex and genetic background (Meneses G et al. PLoS One. 13:e0201375. 2018.). These evidences require further investigation to evaluate whether CDAA/LPS exert similar effect in female mice.

Moreover, as the reviewer suggested, it has been reported that 30 housing potently exacerbates NASH pathogenesis. In our present study, we reared mice under standard condition, because we aimed to simply evaluate the impact of exogenous LPS administration on CDAA-fed NASH. However, considering further enhancement of LPS impact on NASH progression, thermoneutral housing should be a critical booster.

These are certain limitation of our present study. Therefore, we mentioned these aspects in Discussion of the revised manuscript (line 188-192).

The discussion is too long and poorly focused on novel findings. For example Lines 145-162 should better be deleted (This is a topic to be addressed in the Background section). Rework the layout of this section into a four-paragraph structure as follows: a) Start the discussion by summarizing novel findings. b) Next, discuss them in the setting of previous study. c) List limitations and strengths of the model. d) Finally, propose a research agenda specifically based on novel findings.

Answer

We greatly appreciate the reviewer’s instructive suggestion. As the reviewer exactly mentioned, we rewrote and rearranged Discussion part in the revised manuscript (line 147-205).

Expand the translational potential of findings, as much as possible. For example, it must be highlighted that decreased choline intake is significantly associated with increased fibrosis in postmenopausal women with NAFLD (Am J Clin Nutr. 2012;95:892-900) which, again, raised the topic of sex differences in NAFLD.

Answer

We thank for the reviewer’s suggestion. As the reviewer correctly stated, we raised some translational potentials of our findings. Of course, it is important to clinically investigate the association between choline intake and NASH fibrosis in conjunction with endotoxin levels. In line with this, serum concentration of choline may be a biomarker to evaluate severity of NASH. Additionally, it is also required to elucidate whether pharmacological reduction of LPS, by for example non-absorbable antibiotics, inhibits fibrosis development in NASH patients with decreased intake of choline. We added these proposals in Discussion of the revised manuscript (line 192-198).

Round 2

Reviewer 2 Report

I would like to thank these Authors on fully and satisfactorily addressing all the points I have raised. As a result of their constructive attitude, this submission is definitely improved.